# Quantum octets in high mobility pentagonal two-dimensional PdSe$_2$

Yuxin Zhang [1], Haidong Tian[1], Huaixuan Li[2,3], Chiho Yoon[2], Ryan A. Nelson[4], Ziling Li[1], Kenji Watanabe [5], Takashi Taniguchi [6], Dmitry Smirnov [7], Roland K. Kawakami [1], Joshua E. Goldberger[4], Fan Zhang [2] & Chun Ning Lau [1] ✉

Two-dimensional (2D) materials have drawn immense interests in scientific and technological communities, owing to their extraordinary properties and their tunability by gating, proximity, strain and external fields. For electronic applications, an ideal 2D material would have high mobility, air stability, sizable band gap, and be compatible with large scale synthesis. Here we demonstrate air stable field effect transistors using atomically thin few-layer PdSe$_2$ sheets that are sandwiched between hexagonal BN (hBN), with large saturation current > 350 μA/μm, and high field effect mobilities of ~ 700 and 10,000 cm$^2$/Vs at 300 K and 2 K, respectively. At low temperatures, magnetotransport studies reveal unique octets in quantum oscillations that persist at all densities, arising from 2-fold spin and 4-fold valley degeneracies, which can be broken by in-plane and out-of-plane magnetic fields toward quantum Hall spin and orbital ferromagnetism.

Recently, a class of 2D materials with puckered pentagonal lattice structures[1–4] has arrived at scene. These materials have broken sublattice symmetry, and have been predicted to host exotic new properties partly due to the in-plane and cross-plane anisotropy[5–7], including larger band gap variation, axis-dependent conduction polarity, enhanced spin–orbit coupling[8], and nonsymmorphic symmetry-enforced band topology in the monolayer limit[9]. One such example is PdSe$_2$[10–17]. Notably, bulk PdSe$_2$ are reported to display air stability[18], a widely tunable band gap that varies from 0.5 eV in bulk to 1.3 eV for monolayers[1,18–20], ambipolar transport[21], superconductivity upon transforming to a cubic polymorph under high pressure[22], and superior optical and thermoelectric properties[23–25]. Wafer-scale synthesis of few-layer PdSe$_2$ has already been developed[17,26,27]. Mobilities of up to 200 cm$^2$/Vs have been reported, albeit only under a large source-drain bias of 1–2 V[1,18,28]. Thus, as a recent addition to the family of 2D materials, PdSe$_2$ holds great promise for digital electronic, thermoelectric, and optoelectronic applications.

Despite the increasing interest in PdSe$_2$ in the past few years, transport measurements so far have been performed only under high bias and at room temperature, with little optimization with regard to contact resistance or systematic investigation of device behavior. Here, we report transport studies of atomically thin PdSe$_2$ field-effect transistors that are stable in ambient conditions for >1 month, high saturation current among atomically thin semiconductors, as well as high field-effect mobility. This ultra-high mobility enabled the experimental observation of Shubnikov-de Hass oscillation and the quantum hall effect in this pentagonal 2D material. Interestingly, the Landau fan reveals an eightfold degeneracy, arising from twofold spin and fourfold valley degeneracies[29]; increasing in-plane ($B_∥$) and out-of-plane ($B_⊥$) magnetic fields leads to Landau level crossings and broken spin and valley symmetries. Those observations indicate a unique band structure and spin-valley interplay in the pentagonal PdSe$_2$.

Each unit cell of PdSe$_2$ consists of two inversion-symmetric planes of d$^8$ Pd$^{2+}$ ions bonded in square-planar-like coordination with

[1]Department of Physics, The Ohio State University, Columbus, OH 43210, USA. [2]Department of Physics, The University of Texas at Dallas, 800 West Campbell Road, Richardson, TX 75080-3021, USA. [3]Department of Physics, Carnegie Mellon University, Pittsburgh, PA 15213, USA. [4]Department of Chemistry and Biochemistry, The Ohio State University, Columbus, OH 43210, USA. [5]Research Center for Electronic and Optical Materials, National Institute for Materials Science, 1-1 Namiki, Tsukuba 305-0044, Japan. [6]Research Center for Materials Nanoarchitectonics, National Institute for Materials Science, 1-1 Namiki, Tsukuba 305-0044, Japan. [7]National High Magnetic Field Laboratory, Tallahassee, FL 32310, USA. ✉e-mail: lau.232@osu.edu

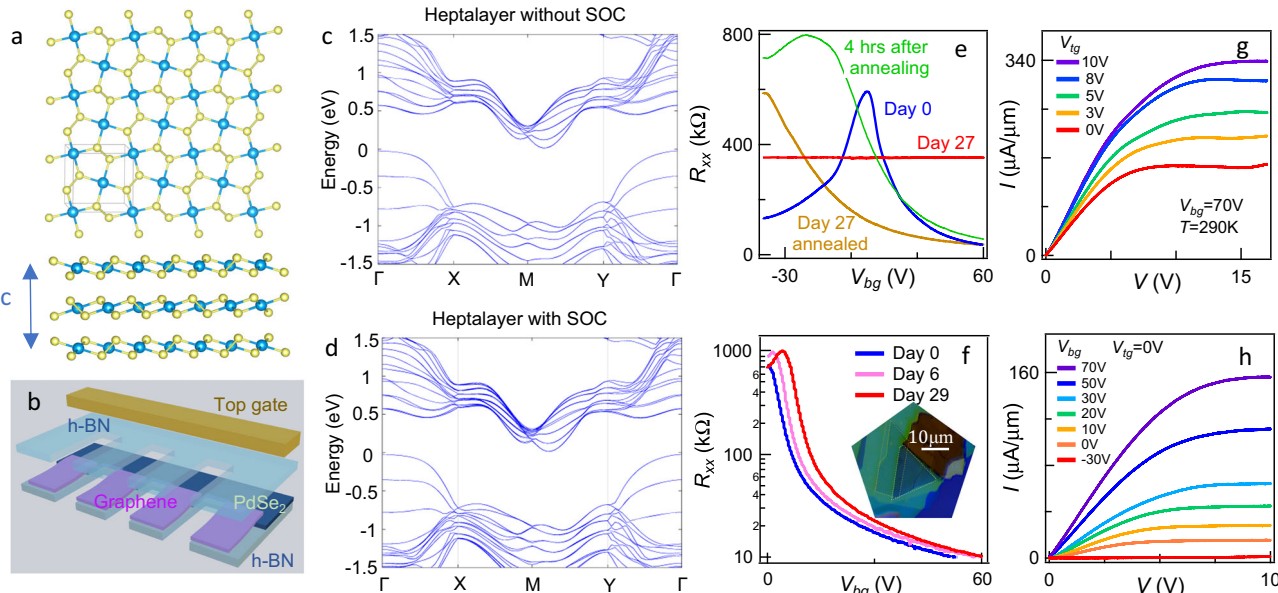

**Fig. 1 | Electronic and room temperature transport properties of few-layer PdSe2 field effect transistors. a** Top and side views of the lattice structure of 2D puckered $PdSe_2$. Blue and yellow spheres represent the Pd and Se atoms, respectively. The vertical lattice constant, $c$ = 7.69 Å. **b** The device schematic. **c, d** Band structures of 7-layer $PdSe_2$ without and with spin-orbit coupling included in the calculation, respectively. Our first-principles calculations based on density functional theory were performed using the Vienna ab initio simulation package (see Supplementary Note 3). **e, f** Transport characteristics $R_{xx}(V_{bg})$ of $PdSe_2$ devices on

$SiO_2$ (device B1) and encapsulated within hBN (device A1), respectively, after periods of time in air. $R_{xx}$ is the four-probe longitudinal resistance, and $V_{bg}$ is the back gate voltage. Insert of **f:** optical image of a hBN/$PdSe_2$/hBN stack. Yellow and white dotted lines outline graphene and $PdSe_2$, respectively. **g, h** I−V characteristics of a 5-layer device A2 at different gate voltages, showing saturation current of ~350 μA/μm when highly doped. **g:** varying top gate voltage $V_{tg}$ at $V_{bg}$ = 70 V. **h:** varying $V_{bg}$ at $V_{tg}$ = 0 V.

diselenide dianions (Fig. 1a, b), yet the two adjacent layers are related by glide-mirror symmetry. The lattice constants are $a$ = 5.74 Å, $b$ = 5.86 Å, and $c$ = 7.69 Å as measured with x-ray diffraction (see Supplementary Information (SI)). Note that the unit cell of the bulk consists of two $PdSe_2$ layers separated by a van der Waals gap; from atomic force microscope (AFM) measurements, each monolayer step has a height of 5.2 Å (see SI). When projected onto a plane, each layer consists of a puckered network of pentagonal rings. Our First-principles calculations show that $PdSe_2$ few-layers are indirect-gap semiconductors, with a 1.06 eV band gap for monolayer and a <0.1 eV band gap for heptalayer (Fig. 1c, d, and see SI). Their conduction band minima are located close to the $M$ point in the Brillouin zone and thus form four symmetry-related valleys. This valley quartet is universal in our few-layer calculations, although the band gap decreases with increasing the thickness as approaching the bulk limit (see SI). (Note that the GW method yields larger band gaps for quasiparticle bands[29].)

## Results and discussion
### Sample characterization at room temperature
Bulk $PdSe_2$ crystals were purchased commercially, or grown via vertical Bridgman inSe flux (see SI), and exfoliated into atomically thin layers onto insulating substrates. Two types of devices are fabricated. We first fabricate "bare" devices on Si/$SiO_2$ substrates, by transferring few-layer graphene on few-layer $PdSe_2$ sheets as electrodes, and depositing Cr/Au contacts on graphene. The blue curve in Fig. 1e displays the four-terminal resistance $R_{xx}$ as a function of back gate voltage $V_{bg}$ for an as-fabricated 4 nm-thick device (device B1). The device appears to be ambipolar, with maximum $R_{xx}$ ~ 600 kΩ located at $V_{bg}$ = 7.8 V. In the linear response regime, its electron and hole field effect mobilities $\mu_{FE}$ = $(1/e)(d\sigma/dn)$ are ~24 and 11 cm²/Vs, respectively (here $\sigma$ is the 2D conductivity, $n$ the charge density, and $e$ the electron charge). To test the device's stability in ambient condition, we leave the device in the air and monitor the transfer curve as a function of time. Surprisingly, despite previous claims of $PdSe_2$'s air stability[1,20,30],

the device deteriorates steadily with time—$\mu_{FE}$ decreases, while the resistance maximum shifts to the right; this $p$-doping aging effect likely arises from oxidation into $PdSe_2O_x$[31]. By day 27, the device loses all response to gate, and $R_{xx}$ = 350 kΩ. We find that thermal annealing at 200 °C in vacuum restores the mobility of the electron-doped regime, though the resistance maximum remains at $V_{bg}$ < −40 V, indicating that the device is electron-doped, which could arise from the formation of Se vacancies. Exposing the annealed device to air for a few hours, we again observe $p$-doping (Fig. 1e, green curve).

To fabricate air-stable, high-performance field effect transistor devices, we take advantage of hexagonal BN (hBN) layers, which have been shown to provide protection for air-sensitive materials such as phosphorene[32], InSe[33], and $CrI_3$[34]. $PdSe_2$ sheets that are 3–8 layers thick are contacted by few-layer graphene leads and sandwiched between hBN layers (Fig. 1b). The global Si/$SiO_2$ back gate tunes both the charge density $n$ of $PdSe_2$ layer and the contact between graphene and $PdSe_2$, while the top gate covers only the channel region and tunes only $n$ therein. Graphene contacts are advantageous over conventional metallic (e.g., Cr/Au or Ti/Au) contacts, since both the work function and surface potential of graphene are gate-tunable, thus the Schottky barrier between graphene and the semiconductor can be effectively lowered. As shown in Fig. 1f, for $V_{bg}$ ≤ 0, as-fabricated hBN-encapsulated devices have high resistance, >~ MΩ (see Fig. S3a); as $V_{bg}$ increases from 0, four-terminal resistance drops rapidly to a few kΩ upon electron doping. The high resistance and low mobility in the $p$-doped regime in these pristine hBN-encapsulated devices suggest that the hole conduction in unencapsulated $PdSe_2$ devices is not intrinsic, but arises from oxidation of $PdSe_2$ that can be partially reversed by vacuum annealing. In contrast to "bare" devices, the hBN-encapsulated devices are very stable in air, with no degradation in mobility and only slight hole doping after 29 days (Fig. 1f).

We now focus exclusively on hBN-sandwiched $PdSe_2$ devices. Figure 1g, h displays the two-terminal current-voltage (I−V) characteristics of a $PdSe_2$ device A2 that is ~2.5 nm thick or ~5 layers at room

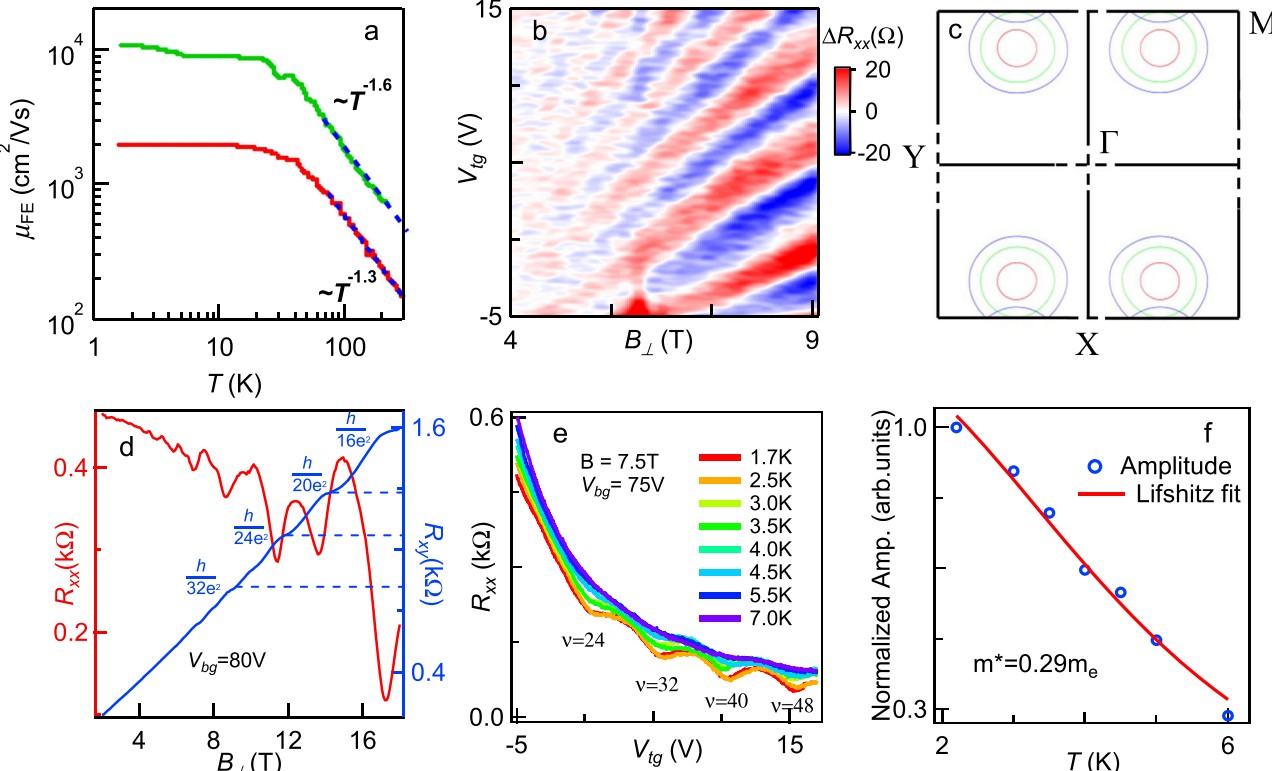

**Fig. 2 | Transport data of PdSe2 transistors at low temperatures. a** Field effect mobility as a function of temperature of two PdSe$_2$ transistors that are ~3.5 nm(green) and 2.5 nm (red) in thickness. Dashed lines are fits to $T^{\gamma}$. **b** SdH oscillations of device A4 versus $V_{tg}$ and the perpendicular magnetic field $B_{\perp}$. To better display the oscillations, a smooth polynomial background is subtracted. **c** Energy contours of conduction band of 7-layer PdSe$_2$ in the $k_x$-$k_y$ plane, showing the four valleys. To better showcase the valleys, segments of the axes are omitted (indicated by dotted lines). The MATLAB function interp2 was used to interpolate the first-principles results and generate the equal-energy contours at specific energies (see Supplementary Note 3). **d** Longitudinal resistance $R_{xx}$ (red curve) and Hall resistance $R_{xy}$ (blue curve) of device A5 plotted versus $B_{\perp}$ at $V_{bg}=80$ V and $T=50$ mK. Inserted values indicate the quantized Hall resistance. **e** $R_{xx}(V_{tg})$ of device A4 at $B_{\perp}=7.5$ T and different temperatures. **f** Normalized amplitude of oscillations in d plotted versus temperature, fitted to the Lifshitz-Kosevich equation.

temperature. When $V_{bg}<0$ V, the device is intrinsic the zero-bias resistance is ~350 MΩ, and saturation current $I_{sat}$ is a few nA/μm. With increasing doping, the two-terminal resistance $R_{2T}$ decreases and $I_{sat}$ increases. At $V_{bg}=70$ V and $V_{tg}=10$ V, $I_{sat}$ ~ 350 μA/μm, which is very high for atomically thin 2D semiconductors. Though $I_{sat}$ per layer is lower than the state-of-the-art value of, e.g. MoS$_2$[35], we note that the saturation current does not necessarily scale with the number of layers in the few-layer limit. Moreover, the channel lengths of our devices are rather long (~1 μm), while the contact resistance is still quite prominent (e.g. when highly doped, $R_{2T}$ and $R_{xx}$ are ~35 kΩ and 7 kΩ, respectively), thus we expect that $I_{sat}$ can be improved by an order of magnitude by minimizing contact resistance of the future generation of devices, for example, via fabricating gates that independently tune the work function of the graphene contacts.

### Observation of quantum octets

To examine the dominant scattering mechanism, we plot the average field effect mobility $\mu_{FE}$ between 4.5 and $6.5 \times 10^{12}$ cm$^{-2}$ as a function of temperature $T$ for two separate devices. As shown by Fig. 2a, the room temperature $\mu_{FE}$ is ~150 and 780 cm$^2$/Vs for the 5-layer (A2) and 7-layer (A3) devices, respectively. At high temperatures, $\mu_{FE}$ increases with decreasing temperature, with a power-law behavior $T^{\alpha}$. Fitting the data yields $\alpha=1.3$ and 1.6 for ~5-layer and 7-layer device, respectively, indicating that the main scattering mechanism is phonon scattering, most likely longitudinal acoustic and longitudinal optical phonons[36]. For $T<30$ K, $\mu_{FE}$ saturates to 2000 and 10,000 cm$^2$/Vs, respectively, indicating that the devices have reached the regime where the mobility bottleneck is scattering

by intrinsic defects and/or impurities. These mobility values, which are 1-3 orders of magnitude higher than prior reports on PdSe$_2$[1,19,23,37] and close to state-of-the-art MoS$_2$ devices[38], suggest that hBN encapsulation significantly reduces the formation of defects and scattering by charged impurities in the substrates.

We now turn to the magnetotransport measurements at the cryogenic temperatures. Here, we focus on a sample that is ~3.5 nm thick or ~7 layers (device A4). Figure 2b shows the background subtracted longitudinal resistance $\Delta R_{xx}$ plotted versus $V_{tg}$ and the perpendicular magnetic field $B_{\perp}$, while the back gate voltage is maintained at $V_{bg}=65$ V. Clear Shubnikov-de Hass (SdH) oscillations from PdSe$_2$ start to be resolved at ~4 T, indicating quantum mobility of 2500 cm$^2$/Vs. Interestingly, the Landau fan features resistance minima at filling factors $\nu=nh/eB_{\perp}$ that are integer multiples of 8 (Fig. 2b, d); such quantum Hall octets that persist at all density ranges have not been observed in other 2D materials. To account for this 8-fold degeneracy, we note that the conduction band bottom occurs near the $M$ point in the rectangular Brillouin zone, and that the glide and screw symmetries dictate a 4-fold valley degeneracy (Fig. 2c). Additionally, near the conduction band bottom, the spin-orbit coupling is negligibly small, as can be seen in Fig. 1c, d and Fig. S6–7, the bands have a 2-fold spin degeneracy even for even-layer systems in which inversion symmetry is broken. Consequently, the charge carriers in PdSe$_2$ are endowed with the observed 8-fold degeneracy. In addition, Hall measurements was performed on a 3-L device A5, as shown in Fig. 2d, quantized Hall resistance $R_{xy}$ plateaus are resolved at $R_{xx}$ minimum, attesting to the quantum Hall nature of the observed octets. Additional transport data on A5 can be found in Supplementary Fig. 3b–d.

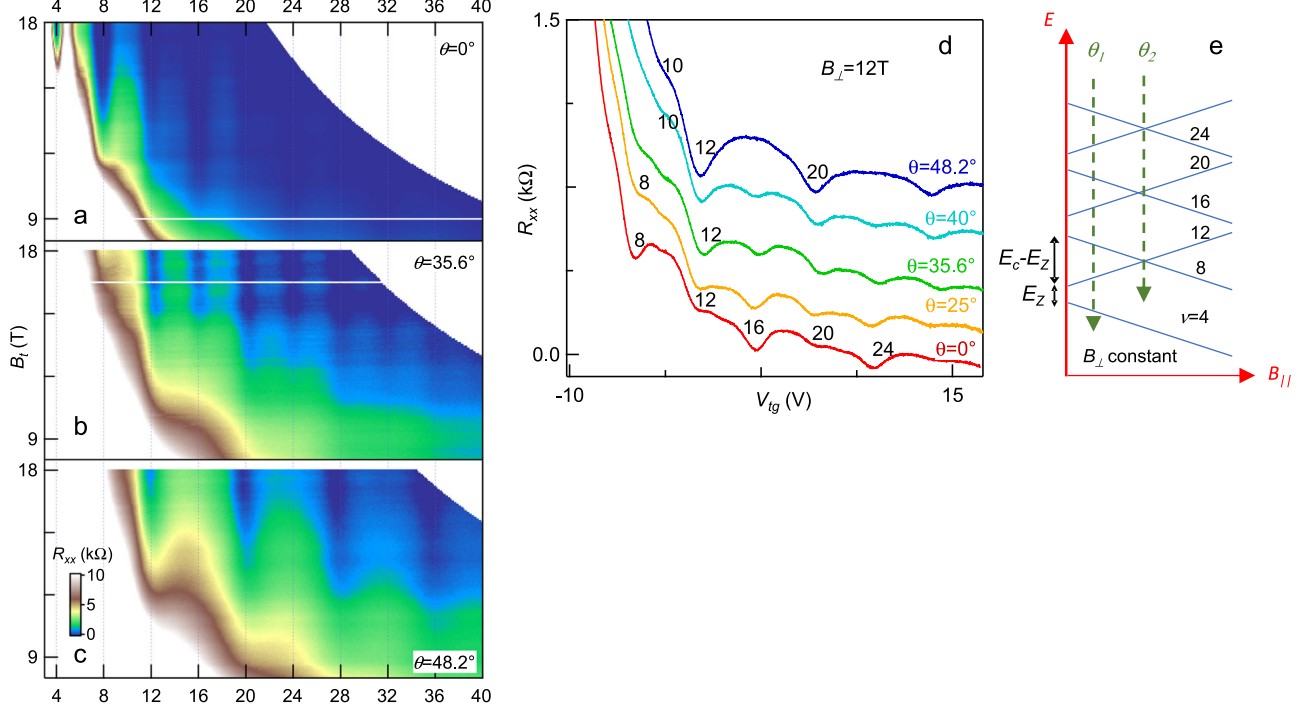

**Fig. 3 | Magnetotransport data in tilted magnetic fields for device A4 at T = 50 mK. a–c** $R_{xx}$ in kΩ versus total magnetic field $B_t$ and filling factor $v$ at different tilting angles. **d** Line traces of $R(V_{tg})$ at $B_\perp$ = 12 T and different tilting angles. The traces are offset for clarity. Black numbers indicate filling factors. **e** Schematics of the evolution of Landau levels with in-plane magnetic field while $B_\perp$ is kept constant. $E_c = \hbar\omega_c$, is the cyclotron gap. $E_Z = g\mu_B B_t$, is the Zeeman gap. $\theta_1$ indicates the angle at which all adjacent states are approximately equally resolved. $\theta_2$ is the coincidence angle at which LLs with opposite spin indices cross.

The effective mass of the charge carriers is extracted by evaluating the amplitude of the SdH oscillations with temperature. Figure 2e plots $R(V_{tg})$ at $T$ ranging from 1.7 K to 7 K at $B_\perp$ = 7.5 T, and Fig. 2f the oscillation amplitude as a function of temperature. Fitting the data to the Lifshitz–Kosevich formula $\Delta R \propto \frac{\chi}{\sinh(\chi)}$, where $\chi = \frac{2\pi^2 kTm^*}{\hbar e B_\perp}$, $k$ is the Boltzman constant, $\hbar$ is the reduced Planck constant, and $T$ is the temperature, we obtain an effective mass $m^* = 0.29 m_e$, here $m_e$ is the bare electron mass in vacuum. This result is in excellent agreement with previous theoretical calculations[29].

## Magnetotransport measurements in tilted fields

At higher fields, degeneracy of the quantum Hall octets is partially lifted (Fig. 3d and Fig. S3b). Major resistance valleys occur at filling factors $v = 8N$, where $N = 1, 2, 3...$ is an integer denoting the Landau level (LL) index; between the major valleys, minor resistance dips are visible at $v = 4N_{odd}$, where $N_{odd} = 1, 3, 5...$ is an odd integer. To gain insight into the nature of this quantum Hall ferromagnetism, we perform measurements in tilted magnetic fields, since the valley (orbital) degrees of freedom depend only on the perpendicular component of $B$, whereas the spin couples to the total magnetic field $B_t$. Figure 3a–c plots the $R(B_t, v)$ for data acquired at angles $\theta = 0, 35.6°$ and $48.2°$, respectively, and line traces of $R(V_{tg})$ are shown in Fig. 3d. At $\theta = 0$ (i.e., $B_t = B_\perp$ and $B_\parallel = 0$), the QH states at $v = 8, 16, 24$, and 32 are resolved at lower fields, and those at $v = 4, 12, 20$ and 28 are resolved at higher fields with shallower minima, indicating that the gaps for the former (latter) are larger (smaller). At $\theta = 35.6°$, the adjacent states are approximately equally resolved, suggesting that the gaps at $v = 8N$ ($v = 4N_{odd}$) decrease (increase) with increasing $B_\parallel$. Upon further increase of $B_\parallel$, at $\theta = 48.2°$, only the QH states at $v = 4N_{odd} = 12, 20, 28$, and 36 are resolved, i.e., the cyclotron and Zeeman gaps become equal in magnitude, and the LLs with indices $(N, \uparrow)$ and $(N + 1, \downarrow)$ cross each other. This is the so-called coincidence angle, at which LLs with opposite spin indices cross.

These data enable us to determine the schematic of LL resolution. At low field, the 8-fold degenerate LLs are separated by cyclotron gaps $E_c = \hbar\omega_c(N + \frac{1}{2})$, where $\omega_c = \frac{eB_\perp}{m^*}$ is the cyclotron frequency. Increasing $B_\perp$ first lifts the spin degeneracy, giving rise to Zeeman-split LLs with energy difference $E_Z = g\mu_B B_t$, where $\mu_B$ is the Bohr magneton and $g$ the effective electron $g$-factor, including the Coulomb interaction correction. In tilted field measurements, increasing $B_\parallel$ further enhances the Zeeman splitting, giving rise to LL crossing at large $B_\parallel$ (Fig. 3e). Using this schematic, we can extract the effective $g$-factor from the coincidence angle at which the cyclotron and Zeeman gaps are equal in magnitude ($\theta = 48.2°$). This condition yields $g = \frac{\hbar e}{\mu_B m^*} \cdot \frac{B_\perp}{B_t} = \frac{2\cos(\theta_2)}{m^*/m_e} = 4.60$, respectively, using $m^* = 0.29 m_e$ obtained earlier. This enhancement from the bare value of $g = 2$ in a nearly spin-orbit-coupling-free system substantiates the presence of electron–electron interaction effect and the resulting quantum Hall spin ferromagnetism.

## Lifting of all degeneracy

Finally, we perform measurements at even higher $B_\perp$ fields up to 29 T and observe that, after the lifting of the spin degeneracy, the fourfold valley degeneracy is also broken (Fig. 4a). This quantum Hall orbital ferromagnetism is clearly illustrated by the emergence of additional resistance minima between filling factor 4, 8, and 12 (Fig. 4b). Future study is required to show whether these quantum Hall states are valley coherent phases breaking translational symmetry or valley polarized phases exhibiting in-plane ferroelectricity[39]. Notably, quantum hall effects down to the lowest LL are resolved, attesting to the excellent quality of our sample. The entire sequence of degeneracy breaking is schematically illustrated in Fig. 4c.

In conclusion, we demonstrate that few-layer pentagonal $PdSe_2$, when sandwiched between hBN layers, is an excellent 2D semiconductor, boasting air stability, high saturation current, exceedingly high field effect mobility, and quantum Hall octets, and ferromagnetism

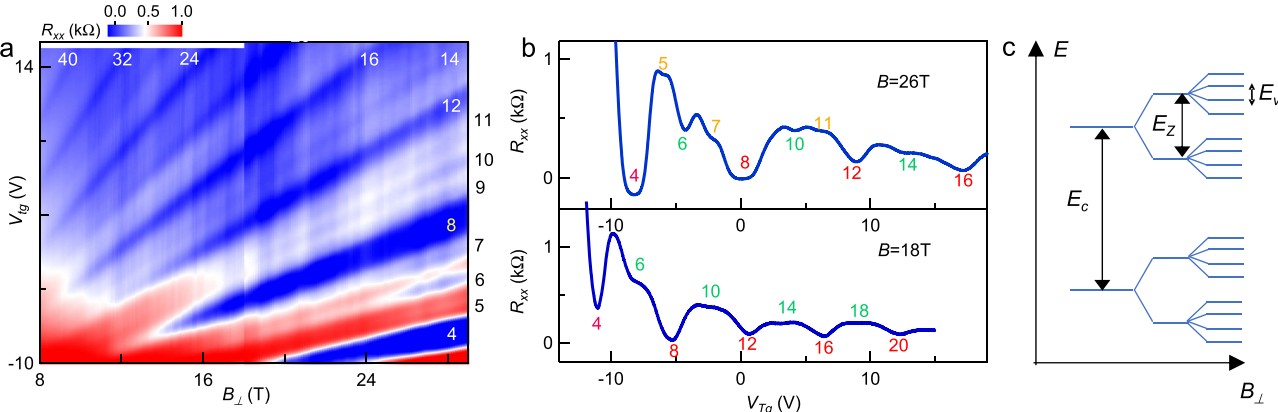

**Fig. 4 | Magnetotransport data at high magnetic fields for device A4 at**
$T$ = 50 mK. a $R_{xx}(V_{tg}, B_\perp)$ in kΩ at $V_{bg}$ = 75 V from $B_\perp$ = 8 to 29 T. b Line traces $R_{xx}(V_{tg})$ at $B_\perp$ = 18 and 26 T, respectively. Filling factors are labeled on the corresponding resistance dips (the negative resistance near ν = 4 is due to an instrumental artifact). c Schematics of the resolution sequence of the cyclotron gap $E_c$, Zeeman gap $E_Z$, and valley gap $E_v$.

in magnetic fields. Our work paves the way for future electronic, photonic, and topological applications of this pentagonal material.

## Methods

### PdSe₂ synthesis

Bulk PdSe₂ crystals were purchased from Sixcarbon Technology or grown via vertical Bridgman-type growth. Elemental Pd powder is mixed with a large excess of elemental Se powder (molar ratio of 3:97 Pd:Se). The typical 7-gram charge is loaded into a fused-silica ampoule with 2–3 mm thick walls and a tapered end. It is important to ensure that even if all of the Se was volatilized, the pressure generated would be much less than the critical hoop stress of fused-silica (<50 MPa). The ampoule is evacuated and sealed under a typical vacuum pressure of ~60 mtorr. After sealing, a small fused-silica hook is welded to the ampoule opposite the tapered end. The ampoule is then suspended by this hook within a vertical single-zone tube furnace using nichrome wire to suspend the ampoule. To grow the PdSe₂ single crystals, the ampoule is suspended in the vertical furnace which is ramped up to 850° C in 24 h. The temperature is held at 850 °C for 50 h before the ampoule is dropped through the natural thermal gradient of the vertical furnace at a rate of 1 mm h⁻¹ over the course of ~100 h. X-ray diffraction was used to confirm the lattice parameters of the bulk, giving $a$ = 5.7423(5) Å, $b$ = 5.8646(5) Å, c = 7.6924(2) Å.

### Sample characterization and device fabrication

Thin layer flakes were exfoliated on PDMS using the standard Scotch tape method. The thickness of thin flakes was measured using an atomic force microscope (AFM). For hBN-encapsulated devices and bare devices, hBN/graphene/PdSe₂/hBN or graphene/PdSe₂/hBN stacks were assembled using a dry transfer technique with a PC stamp[40], then released onto 290 nm SiO₂/Si substrates at 170°C ~ 180 °C. Afterwards, the stacks were shaped to the desired Hall bar structure through electron-beam lithography and reactive ion etching (RIE) with SF₆ and Ar, followed by the deposition of top gate dielectric (Al₂O₃ 50-70 nm). Finally, top gate and metal leads that make edge-contact to graphene were patterned by electron-beam lithography and subsequent deposition of metals (Cr 5 nm /Au 100 nm).

### Electrical and magnetotransport measurements

Two-terminal transport characteristics were measured by applying an a.c. voltage bias of 50–100 μV (a standard lock-in amplifier SR830) to the source and recording the drain current. For four-terminal measurements, the SR830 was used to apply a constant a.c current of 10 to 100 nA and measure the voltage drop across the channel. Top gate and back gate d.c bias were applied using Keithley 2400.

Magnetotransport measurements were performed in a Janis cryostat and two other He3 cryostats at the National High Magnetic Field Laboratory (NHMFL).

## Data availability

The data that support the findings of this study are available within the main text and Supplementary Information. Any other relevant data are available from the corresponding authors upon request. Source data are provided with this paper.

## Code availability

The codes for band structure calculations are available upon request from F.Z. and H.L.

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

## Acknowledgements

We thank Dima Shcherbakov for the helpful discussion. This work is supported by NSF/DMR 2128945. H.T. is supported by the DOE BES Division under grant no. DE-SC0020187. R.A.N. and J.E.G. acknowledge the Air Force Office of Scientific Research for funding from grant number FA9550-21-1-02684. Z.L. and R.K.K. acknowledge support from AFOSR/MURI project 2DMagic (FA9550-19-1-0390) and the US Department of Energy (DE-SC0016379). A portion of this work was performed at the National High Magnetic Field Laboratory, which is supported by the National Science Foundation through NSF/DMR-1644779 and the State of Florida. K.W. and T.T. acknowledge support from the JSPS KAKENHI (Grant Numbers 20H00354, 21H05233, and 23H02052) and World Premier International Research Center Initiative (WPI), MEXT, Japan. The theoretical work done at UT Dallas was supported by NSF under Grants no. DMR-1945351, no. DMR-2105139, and no. DMR-2324033. We acknowledge the Texas Advanced Computing Center (TACC) for providing resources that have contributed to the research results reported in this work.

## Author contributions

C.N.L. conceived the experiment. Y.Z. fabricated the devices. Y.Z., H.T., and D.S. performed measurements. Z.L and R.K. performed second harmonic generation measurements. R.A.N. and J.E.G. grew PdSe2 bulk crystals. K.W. and T.T. provided hBN crystals. H.L. and C.Y. performed the theoretical calculations under the supervision of F.Z. C.N.L. and Y.Z. analyzed data. C.N.L, Y.Z., and F.Z. interpreted data and wrote the manuscript. All authors read and commented on the manuscript.

## Competing interests

The authors declare no competing interests.
