## [Peer Review File · Nature Communications]

Quantum Octets in High Mobility Pentagonal Two-Dimensional PdSe₂REVIEWER COMMENTS

Reviewer #1 (Remarks to the Author):

Yuxin Zhang and co-authors report an electrical transport study of few-layer PdSe₂. They build high-quality transport devices and perform measurements as a function of temperature, magnetic field, and doping. I believe that this manuscript is timely because interest in non-hexagonal 2D materials (such as PdSe₂) is rapidly growing in the community. Before suggesting this paper for publication I have several questions and comments:

1) Authors report saturation current in their transistors, the value is 350 $\mu\text{A}/\mu\text{m}$. Despite this value being high and approaching some of the highest values reported in the literature for transistors built out of 2D materials, my concern is that they are reporting it for multilayer devices (in that case, 5-layer device). They are comparing this value with literature. A lot of literature reports similar currents for monolayers. My suggestion is to report the current density per layer rather than the current density per 5-layer device. This will make comparisons more fair and will allow us to have more universal metrics for understanding state of the art for transistors made of 2D materials.

2) I am also a little bit concerned about the title. The title "Quantum Octets in Air Stable High Mobility Two-Dimensional PdSe₂" implies that PdSe₂ is air-stable. Reading the manuscript I can see that it is not air-stable, which is explained on page 2. I understand that authors are trying to highlight the fact that they are measuring the intrinsic properties of this material by protecting it with BN but this doesn't mean that PdSe₂ is air-stable. I strongly suggest rephrasing the title, otherwise, it will be highly misleading.

The next questions are related to low-temperature quantum oscillation study and I think that clarifying my concerns will be crucial in understanding if this study represents intrinsic properties of PdSe₂ or PdSe₂ + graphite oscillations. Many of the claims of the paper, specifically on the valley and spin degeneracy of PdSe₂ rely on quantum oscillations interpretation.

3) The manuscript doesn't discuss, at least I didn't see a clear explanation of how the authors disentangled QO from graphite and from PdSe₂. How did authors make sure that graphite contacts don't contribute to the observed oscillations and if it does, how did they remove its contribution (if it is possible to remove such contribution at all)?

4) Did the authors investigate even-odd layer dependence of quantum oscillations?

5) Is the valence band and QO accessible at low temperatures?

6) What is the effect of an electric field on observed behavior? It seems that the authors have dual gate geometry that could allow them to perform such measurements.

Reviewer #2 (Remarks to the Author):

The work by Zhang et al. describes several significant advances in the field of 2D materials. They explore a few-layered PdSe₂, a 2D semiconductor that was studied much less compared to its more famous

cousins, such as MoS₂. First, the authors create high-quality hBN-encapsulated PdSe₂ devices with carrier mobility much higher than what was reported previously. The mobility is so high as to be of interest in the context of semiconductor technology. Second, their temperature-dependent data leads to insight into scattering mechanisms in this novel material. Finally, the high device quality allows the authors to study for the first time the Quantum Hall effect physics in PdSe₂. They observe a novel transport signature – 8-fold degenerate Landau levels that are broken into lower degeneracy at the high magnetic field. This degeneracy allows drawing conclusions regarding the material's bandstructure.

The results are novel and very interesting. They surely are going to lead to an uptick of interest in this novel 2D material. The data is of high quality and definitely worth publishing. I wholeheartedly recommend the publication of this manuscript.

I only have two minor comments:

- 1) It would be worthwhile to comment more on the scattering mechanisms. Can the authors identify phonon-assisted mechanisms leading to T-dependent scattering? Why does mobility saturate at T<30K?
- 2) Were Hall measurements done, and was R_{XY} measured? If yes, it would be worthwhile to compare Hall mobility to field effect mobility and show complementary QH data. If not, it probably should be explained.
- 3) A comment on the role of contacts would be in order. What contact material may improve interface resistance?
- 4) In the sentence on line 161, there should be a comma instead of a period.

Reviewer #3 (Remarks to the Author):

Zhang et al present work on the transport properties of PdSe₂. The material is of interest to the community due to the possibility of enhanced stability and mobility compared to other 2D materials. The results are generally of interest, but the data and the presentation are not sufficient to justify the claims of the manuscript and I can therefore not recommend the manuscript for publication in its current form. In particular:

- The authors need to define NMDCs

- Higher saturation currents can be observed in MoS₂. The statement on the record high saturation among atomically thin semiconductors needs to be removed or clarified.

<https://doi.org/10.1038/s41586-021-03472-9>

- "Record high field effect mobility" needs to be clarified, is this for all 2D semiconductors and all temperatures? Higher examples are found elsewhere at low temperature.

<https://www.nature.com/articles/s41467-019-08629-9>

- Why do the authors not show the field effect characteristics at negative gate voltages? It is clearly possible, as shown Figure 1e. This needs to be included to remove any doubt of the results. How do we know that the devices have a high resistance for $V_{bg} < 0$ as the authors state if it is not shown? If this is the case, why is the hole mobility so significantly degraded compared to the unencapsulated devices?

- The main text states that the thickness is ~ 5 layers and the figure caption for Figure 1gh says 6 layers. This needs to be consistent.

- How is the "background" subtracted from the ΔR_{xx} plots in 2b? Is the same process used for the amplitude of oscillations?

- How are the filling factors assigned if there is no Hall resistance for confirmation? Could the assignment not be equally valid for multiples of 4?

- Octets have been observed in other 2D systems:

<https://doi.org/10.1038/nphys2008>

<https://doi.org/10.1038/s41586-021-03849-w>

the statement saying it is unique to PdSe₂ should be removed or clarified.

- The authors do not observe the quantum Hall effect as they have no measurement of the Hall resistance or even well defined zero resistivity states. Claims such as "the first experimental observation of... the quantum hall effect" should be removed. "Hall" should be capitalised throughout.

-The inset in Figure 2b needs to be removed as it is covering half of the data.

-There is no scale or units, or even indications on what is being measured in the colour scale in 2b, 3a and 4b. What is "Low" and "High". The resistance values, if that's what they are on the colour scale in Figure 3a only go from 0 to 100hms. This is inconsistent with the rest of the data in the manuscript. The colour bar in 4b only goes to "1". 1 what?

- Why in Figure 3d and 4b are the resistance values negative? This is particularly troubling as the authors mention above that they have subtracted a "background" resistance in previous figures. It would suggest that an unknown background has been subtracted from the curves in 3d and 4b. This needs to be clarified.

Response to Reviewer 1

We thank the referee for careful reading and thoughtful critique of the manuscript. We have carefully revised the manuscript to address the comments and concerns.

1. The referee comments that: “*Authors report saturation current in their transistors, the value is 350 $\mu\text{A}/\mu\text{m}$. Despite this value being high and approaching some of the highest values reported in the literature for transistors built out of 2D materials, my concern is that they are reporting it for multilayer devices (in that case, 5-layer device). They are comparing this value with literature. A lot of literature reports similar currents for monolayers. My suggestion is to report the current density per layer rather than the current density per 5-layer device. This will make comparisons more fair and will allow us to have more universal metrics for understanding state of the art for transistors made of 2D materials.*”

We thank the referee for raising this point. We note that, in the few-layer limit, an N -layer device has N subbands, while we are doping only the first subband; thus the saturation current does not necessarily scale linearly with number of layers. Nevertheless, we agree that the conventional metric should be saturation current density per layer. In response, we have modified the text to read

“At $V_{bg}=70\text{V}$ and $V_{tg}=10\text{V}$, $I_{sat} \sim 350 \mu\text{A}/\mu\text{m}$, which is very high for atomically thin 2D semiconductors. Though I_{sat} per layer is lower than the state-of-the-art value of, e.g. MoS_2 [35], we note that the saturation current does not necessarily scale with the number of layers in the few-layer limit. Moreover, the channel lengths of our devices are rather long ($\sim 1 \mu\text{m}$), while the contact resistance is still quite prominent (e.g. when highly doped, R_{2T} and R_{xx} are $\sim 35 \text{ k}\Omega$ and $7 \text{ k}\Omega$, respectively), thus we expect that I_{sat} can be improved by an order of magnitude by minimizing contact resistance of the future generation of devices, for example, via fabricating gates that independently tune the work function of the graphene contacts.”

2. The referee comments that: “*The title ‘Quantum Octets in Air Stable High Mobility Two-Dimensional PdSe2’ implies that PdSe2 is air-stable. Reading the manuscript I can see that it is not air-stable, which is explained on page 2. I understand that authors are trying to highlight the fact that they are measuring the intrinsic properties of this material by protecting it with BN but this doesn't mean that PdSe2 is air-stable. I strongly suggest rephrasing the title, otherwise, it will be highly misleading.*”

We thank the referee for the comment. On page 2, we meant to show that bare PdSe₂ without encapsulation is sensitive to air, contrary to many reports in the literature. However, such oxidation occurs relatively slowly and can partially reversed by annealing, with electron mobility restored. In this sense, it is more stable than many other 2D materials, such as black phosphorus, CrI₃ and WTe₂.

Nevertheless, we agree with the referee that the original title might be misleading. We now have changed the title to “Quantum Octets in High Mobility Pentagonal Two-Dimensional PdSe₂”

3. The referee asks that “How did authors make sure that graphite contacts don't contribute to the observed oscillations and if it does, how did they remove its contribution (if it is possible to remove such contribution at all)?”

We understand the referee’s concern. We have ascertained that the observed oscillations are intrinsic properties of PdSe₂ without any contribution from the few-layer graphene contacts, based on the following observations:

First, the top gate covers only the PdSe₂ channel but not the graphene-contacted leads, thus it tunes *only* the carrier density in PdSe₂. Therefore, any response to V_{tg} arises only from PdSe₂. Meanwhile, since graphene contacts are on top of PdSe₂, the back gate also has limited tunability on the carrier density in graphene due to screening once PdSe₂ is doped.

Second, all data presented in this manuscript are from four-probe measurements, which eliminate the contact resistance and therefore any contribution from the few-layer graphene contacts. In fact, in two-terminal measurements, few-layer graphene gives rise a series of gate-independent strong oscillations that onsets at small field (<1T) due to its high mobility, obscuring features from PdSe₂ (Fig. S2 in Supplementary Information, reproduced here).

Third, at low fields, quantum oscillations from PdSe₂ at all densities are 8-fold degenerate; in contrast, in the absence of broken symmetries, quantum oscillations in few-layer graphene are 4-fold degenerate, with the only exception of the so-called zero-energy Landau levels, which can produce a $4L$ -fold degeneracy (here L is the number of layers). This special orbital degeneracy of the lowest L Landau levels near the charge neutrality point is evidently not the case for PdSe₂.

In response to the referee’s comment, we have included the above discussion in the Supplementary Information.

4. The referee asks that “Did the authors investigate even-odd layer dependence of quantum oscillations?”

Indeed this is an interesting question, and is part of our on-going work. The low temperature transport data presented in this manuscript are all from PdSe₂ devices with odd number of layers. We are working on collecting transport data from the even-numbered layers PdSe₂ to see if there are distinct quantum oscillation behaviors, which will be presented in a follow-up manuscript.

From our theoretical perspective, however, the even-odd layer effect might turn out to be rather small, despite the broken inversion symmetry in even-layers. This is because the spin-orbit-coupling induced band splitting is negligible at their conduction band bottom, as revealed by our first-principles calculations (see Fig. S7), unlike the case of WSe₂, WS₂, MoSe₂, and MoS₂.

5. The referee asks that “Is the valence band and QO accessible at low temperatures?”

Unfortunately, we find that the hole mobility of hBN-encapsulated devices is very low ($\sim 2 \text{ cm}^2/\text{Vs}$, see figure here), hence no quantum oscillation is observed for the valance band. We believe that this is due to the larger electrical field required to tune the Fermi-level to the valance band, the much heavier effective mass for the Γ -valley holes (see Fig. S7), and that the mobility of holes in PdSe₂ is smaller than that of electrons. The hole conduction observed in prior reports using unencapsulated PdSe₂ devices (e.g. ref. 1 and 37) is likely to be extrinsic, arising from formation of PdSe₂O_x (see, e.g. ref. 31).

In response to the comment, we have included the above figure as Fig. S3a in the Supplement.

6. The referee asks that “*What is the effect of an electric field on observed behavior? It seems that the authors have dual gate geometry that could allow them to perform such measurements.*”

We have indeed examined the devices’ dependence on electric field, which appears to be modest at best. This is briefly discussed in the supplement (Fig. S5 and associated discussion): “*The device exhibits weak dependence on the out-of-plane displacement field, e.g. the quantum Hall states at $\nu = 8$ and 14 appears to be better resolved at larger displacement field, though further systematic studies will be necessary to ascertain the effect.*”

From our theoretical perspective, the out-of-plane displacement field neither breaks the spatial symmetry that yields the 4-fold valley degeneracy nor the time-reversal symmetry that dictates the 2-fold spin degeneracy. This appears to be consistent with our experimental observation.

In response to the comment, we have added the above discussion to SI.

Response to Reviewer 2

We appreciate the referee for considering our results “*novel and very interesting*” recommending the publication of our manuscript. We also thank the referee for their thoughtful critique of the manuscript. We have carefully improved the manuscript to address the comments and concerns.

1. The referee comments that “*It would be worthwhile to comment more on the scattering mechanisms. Can the authors identify phonon-assisted mechanisms leading to T-dependent scattering? Why does mobility saturate at T<30K?*”

The device mobility, given by $\left[\frac{1}{\mu_{imp}} + \frac{1}{\mu_{ph}(T)}\right]^{-1}$, is limited by the dominant scattering mechanism. Here μ_{imp} is the impurity/defects limited mobility, which is typically independent of temperature, and $\mu_{ph}(T) \sim T^{-\alpha}$ is the phonon-limited mobility, which has a power-law dependence on temperature. At sufficiently high temperatures, the abundant thermally excited phonons dominate the scattering process. From studies of 2D transition metal dichalcogenides, the most likely types of phonons that contribute to scattering are the longitudinal acoustic and optical phonons. As T decreases, the decreasing phonon population leads to fewer electron-phonon scatterings, thus μ increases steadily, until at certain crossover temperature T^* where impurity scattering becomes the dominant process. Since the number of impurities/defects is constant in a given device, mobility is independent of temperature for $T < T^*$. The exact value of T^* is determined by the relative strengths and the nature of the scattering processes, e.g. the types of phonons (acoustic, optical, polar, flexural, etc) and the types of impurities (charged/vacancies/dislocation, long/short range etc), and is difficult to identify without extensive experimental and theoretical studies at multi-scales.

In response to the referee’s question, we have revised the text to read

*“Fitting the data yields $\alpha = 1.3$ and 1.6 for ~ 5 -layer and 7 -layer device, respectively, indicating that the main scattering mechanism is phonon scattering, **most likely longitudinal acoustic and longitudinal optical phonons**. For $T < 30K$, μ_{FE} saturates to $2,000$ and $10,000$ cm^2/Vs , respectively, indicating that the devices have reached the regime where the mobility bottleneck is scattering by **intrinsic defects and/or impurities**[36]. These mobility values, which are 1 - 3 orders of magnitude higher than prior reports[1,19,23,37] and close to state-of-the-art MoS_2 devices[38], suggest that **hBN encapsulation significantly reduces formation of defects and scattering by charged impurities in the substrates.**”*

2. The referee comments that “*Were Hall measurements done, and was R_{XY} measured? If yes, it would be worthwhile to compare Hall mobility to field effect mobility and show complementary QH data. If not, it probably should be explained.*”

We thank the referee for this important comment. It is technically challenging to put graphene contacts on both sides of the $PdSe_2$ flakes due to their relatively small size, thus in the original

manuscript we were only able to obtain R_{xx} data. Recently we have obtained R_{yy} data, which have been added to Supplementary Information (Fig. S3d-e).

3. The referee comments that “*A comment on the role of contacts would be in order. What contact material may improve interface resistance?*”

We thank the referee for this comment. We have tried Cr/Au and Ti/Au contacts, both yielding high contact resistance. Graphene contacts are effective since its work function and hence the Schottky barrier at the graphene-semiconductor interface can be tuned by gate. In the future, the contact resistance can be further optimized by fabricating gates that can independently tune the work function of the contacts.

In response, we have added the following sentence to the first paragraph on page 3

“Graphene contacts are advantageous over conventional metallic (e.g. Cr/Au or Ti/Au) contacts, since both the work function and surface potential of graphene are gate-tunable, thus the Schottky barrier between graphene and the semiconductor can be effectively lowered.”

and

“ I_{sat} can be improved by an order of magnitude by minimizing contact resistance of the future generation of devices, for example, via fabricating gates that independently tune the work function of the graphene contacts.”

4. The referee comments that “*In the sentence on line 161, there should be a comma instead of a period.*”

We thank the referee for the careful reading of the manuscript, and have corrected it.

Response to Reviewer 3

We appreciate the referee for considering our work generally interesting. We also thank the referee for careful reading and thoughtful critique of the manuscript. We have carefully revised the manuscript to address the comments and concerns.

1. The referee comments that “*The authors need to define NMDCs.*”

We thank the referee for noticing this omission. We have now changed “NMDC” to “PdSe₂”.

2. The referee comments that “*Higher saturation currents can be observed in MoS₂. The statement on the record high saturation among atomically thin semiconductors needs to be removed or clarified. <https://doi.org/10.1038/s41586-021-03472-9>”*

We thank the referee for pointing this out. We agree that our saturation current is lower than that of the state-of-the-art values of monolayer MoS₂. We now revised the sentence in the abstract to “*large saturation current >350 μA/μm, and record high field effect mobilities 700 and 10,000 cm²/Vs at 300K and 2K*”

and revised the discussion in the text with citation to the reference

“At $V_{bg}=70V$ and $V_{tg}=10V$, $I_{sat} \sim 350 \mu A/\mu m$, which is very high for atomically thin 2D semiconductors. Though I_{sat} per layer is lower than the state-of-the-art value of, e.g. MoS₂[35], we note that the saturation current does not necessarily scale with the number of layers in the few-layer limit. Moreover, the channel lengths of our devices are rather long ($\sim 1 \mu m$), while the contact resistance is still quite prominent (e.g. when highly doped, R_{2T} and R_{xx} are $\sim 35 k\Omega$ and $7 k\Omega$, respectively), thus we expect that I_{sat} can be improved by an order of magnitude by minimizing contact resistance of the future generation of devices, for example, via fabricating gates that independently tune the work function of the graphene contacts”

3. The referee comments that “*Record high field effect mobility” needs to be clarified, is this for all 2D semiconductors and all temperatures? Higher examples are found elsewhere at low temperature. <https://www.nature.com/articles/s41467-019-08629-9>”*

We meant record high mobility in PdSe₂ at all temperatures. To clarify this, we cited the reference and revised the manuscript to read

“These mobility values, which are 1-3 orders of magnitude higher than prior reports on PdSe₂ [1, 19, 23, 36] and close to state-of-the-art MoS₂ devices[37], suggest that hBN encapsulation significantly reduces formation of defects and scattering by charged impurities in the substrates.”

4. The referee asks that “*Why do the authors not show the field effect characteristics at negative gate voltages? It is clearly possible, as shown Figure 1e. This needs to be included to remove any doubt of the results. How do we know that the devices have a high resistance for $V_{bg} < 0$ as the authors state if it is not shown? If this is the case, why is the hole mobility so significantly degraded compared to the unencapsulated devices?*”

We thank the referee for this question. We find that, for hBN-encapsulated devices, the hole mobility is much lower than the electron mobility (see graph to the left). Thus, no quantum oscillations are resolved and we chose to focus only on the electron-doped regime.

As for the “*increased*” hole mobility of unencapsulated devices, we believe that this is not the intrinsic behavior of PdSe_2 – rather, it arises from the oxidation of PdSe_2 into PdSe_2O_x in ambient conditions (more specifically, activated chemisorption of O_2 at Se vacancy site, see, e.g. ref. 31). Thus our hBN-

encapsulated devices display the intrinsic properties of PdSe_2 .

In response to the referee’s comment, we have now included the above figure as Fig. S3a in Supplementary Information, added a curve that shows enhanced hole conduction of annealed unencapsulated devices after exposing to air (green curve in Fig. 1e), and also added the following sentence to the text:

“The high resistance and low mobility in the p-doped regime in these pristine hBN-encapsulated devices suggest that the hole conduction in unencapsulated PdSe_2 devices is not intrinsic, but arises from oxidation of PdSe_2 that can be partially reversed by vacuum annealing.”

5. The referee comments that “*The main text states that the thickness is ~ 5 layers and the figure caption for Figure 1gh says 6 layers. This needs to be consistent.*”

We thank the referee for noticing this typo. We have corrected Fig. 1 caption. In addition, we have also labeled devices and list device information in Table S1.

6. The referee asks that “*How is the “background” subtracted from the ΔR_{xx} plots in 2b? Is the same process used for the amplitude of oscillations?*”

Only data in Fig. 2b have its background subtracted (we fitted each line trace $R_{xx}(V_{bg})$ to a smooth polynomial, which is subtracted from the data). The data shown in Fig. 2c are raw data, that is, the amplitudes of oscillations are determined without any extra processing.

In response to this comment, we now state in the Fig. 2b caption “*To better display the oscillations, a smooth polynomial background is subtracted*”.

7. The referee asks that “How are the filling factors assigned if there is no Hall resistance for confirmation? Could the assignment not be equally valid for multiples of 4?”

The filling factors are calculated from the equation $\nu = \frac{nh}{Be}$, where n is the charge density and B is the out-of-plane magnetic field. n is calculated using geometric consideration of the capacitance between graphene and the gate, as well as the classical Hall resistance when possible. (see also reply to #9).

We respectfully disagree that, at low fields, the assignment is valid for multiples of 4 instead of multiples of 8. First, this would suggest our estimate for charge density, which is based on the geometric capacitance between the gate and the sample, is off by a factor of 2, which would greatly exceed the error bar (estimated at <5%). Second, since we are able to resolve all 8-fold LLs at high fields, the hypothesis of 4-fold degeneracy at low fields would imply the presence of half-integer fractional states at high fields (which would be very interesting!). Third, our first-principles calculations show that the conduction band edge has a 4-fold valley degeneracy, a 2-fold spin degeneracy, and negligible spin-orbit couplings. These features agree well with our experimental observation of 8-fold degeneracy at low fields.

8. The referee comments that “Octets have been observed in other 2D systems: <https://doi.org/10.1038/nphys2008>; <https://doi.org/10.1038/s41586-021-03849-w>; the statement saying it is unique to PdSe2 should be removed or clarified.”

We thank the referee for the comment. The octet mentioned in the first reference occurs in bilayer graphene at the charge neutrality point, due to the 2-fold spin, 2-fold valley, and an approximate 2-fold orbital degeneracy of the $N=0$ and $N=1$ Landau levels. In the second reference co-authored by one of us, the octet refers to eight different quantum anomalous Hall states that exhibit Chern number ± 2 at zero magnetic field. In contrast, the octets in PdSe₂ is an intrinsic property inherited from the four-fold valley degeneracy in addition to the spin degeneracy and persist to all densities.

To clarify this point, we have revised the manuscript to read

“magnetotransport studies reveal unique octets in quantum oscillations *that persist at all densities*”

and

“such quantum Hall octets *that persist to all density ranges* have not been observed in other 2D materials”.

9. The referee comments that “The authors do not observe the quantum Hall effect as they have no measurement of the Hall resistance or even well defined zero resistivity states. Claims such as “the first experimental observation of... the quantum hall effect” should be removed. “Hall” should be capitalised throughout.”

To address the referee’s concern, we have added the following data on the new sample A6 to the supplementary information (Fig. S3b-e, reproduced below).

Fig. S3b shows a line trace at $B=18\text{T}$ from Fig. S3b, the resistance at $\nu=8$ reaches zero. We note that this is raw data, without any subtraction. Fig. S3d presents the $R_{xy}(V_{bg}, B)$ plot. As shown in

Fig. S3e, quantized Hall resistance plateaus are resolved at the longitudinal resistance minimum, attesting that we have indeed observed the quantum Hall effect.

Fig. S3. (a) Field effect characteristic of device A2 at room temperature from -60V to +65V. (b) $R_{xx}(V_{bg}, B)$ of a 3-L device A6. (c) Line trace of $R_{xx}(V_{bg})$ at $B=18T$ from (b). (d) $R_{xy}(V_{bg}, B)$ of A6. The numbers indicate the filling factors. (e) Red (blue) line: $R_{xx}(R_{xy})$ versus B line cut at $V_{bg}=80V$. B is the perpendicular field, $T=50mK$.

10. The referee comments that “The inset in Figure 2b needs to be removed as it is covering half of the data.”

In response to the comment, we have moved the inset out of Fig. 2b and made it into a new panel (Fig. 2c).

11. The referee comments that “There is no scale or units, or even indications on what is being measured in the colour scale in 2b, 3a and 4b. What is “Low” and “High”. The resistance values, if that’s what they are on the colour scale in Figure 3a only go from 0 to 10Ohms. This is inconsistent with the rest of the data in the manuscript. The colour bar in 4b only goes to “1”. 1 what?”

We apologize for the omission of scales. The color bar of Fig. 2b is now labeled ΔR_{xx} (Ω), with range $\pm 20 \Omega$. We have also updated the captions of Fig. 3a and Fig. 4a to indicate that resistance labeled in the color bars is in units of $k\Omega$.

12. The referee asks that “*Why in Figure 3d and 4b are the resistance values negative? This is particularly troubling as the authors mention above that they have subtracted a “background” resistance in previous figures. It would suggest that an unknown background has been subtracted from the curves in 3d and 4b. This needs to be clarified.*”

We thank the referee for raising this question. The traces in Fig. 3d are offset for clarity, and the resistance values are never negative. We have now updated the figure caption to indicate such.

For Fig. 4b, the data presented are raw data without background subtraction. The negative resistance recorded near $\nu=4$ is an artifact arising from the large dynamic range on the lock-in amplifier, and in regions where the signal undergoes abrupt changes. We have observed that this effect can be avoided by using higher gain and slower gate sweeping rate. Since the data in Fig. 4 are only used to qualitatively determine the resolution sequence of the spin and valley degeneracies (in fact, R_{xx} at $\nu=4$ is not needed for such determination), the negative resistance values, while not desirable, do not affect the message of the figure (i.e. Zeeman gaps is resolved before the valley gaps).

In response to the comment, we have added the sentence “*(the negative resistance near $\nu=4$ is due to an instrumental artifact)*” to the figure caption. We have also added data from another sample A6 (Fig. S3), showing better defined resistance minima.

REVIEWERS' COMMENTS

Reviewer #1 (Remarks to the Author):

I thank the authors for answering my questions, as well as clarifying important concerns raised by other referees. I believe that the paper has been significantly improved since the previous submission. I thus recommend this paper for publication in Nature Communications.

Reviewer #2 (Remarks to the Author):

All of my (minor) concerns have been successfully addressed. I now recommend acceptance.

Reviewer #3 (Remarks to the Author):

The authors have improved the manuscript, and it is likely to be publishable. I have a couple of additional comments for the consideration of the authors.

Labels should be included for the legends of figure 1g and h. Is Figure 1h varying the back gate at constant V_{tg} , or vice versa? It's not clear.

Although the captions have been updated referring to the colour scales, it would be clearer if a label could be included in figure 3c, and 4a (and S5).

The "Additional Transport Data" from device A2 in the SI is welcome. I would think the manuscript would be improved if the Hall effect data, such as S3e, were included in the main manuscript.

Response to Reviewer #3

We thank the referee for considering our manuscript improved and providing additional thoughtful critique. We have further revised the manuscript to address the comments.

1. The referee comments that “*Labels should be included for the legends of figure 1g and h. Is Figure 1h varying the back gate at constant V_{tg} , or vice versa? It's not clear.*”

We thank the referee for noticing this omission. We have added labels for the legends of figure 1g and h, also added the following sentence to Fig.1 caption:

“g: varying top gate voltage V_{tg} at $V_{bg}=70V$. h: varying V_{bg} at $V_{tg}=0V$.”

2. The referee comments that “*Although the captions have been updated referring to the colour scales, it would be clearer if a label could be included in figure 3c, and 4a (and S5).*”

In response to this comment, we have added labels for the color scales in Fig 3c, Fig 4a and Supplementary Fig 5

3. The referee comments that “*The "Additional Transport Data" from device A2 in the SI is welcome. I would think the manuscript would be improved if the Hall effect data, such as S3e, were included in the main manuscript.*”

We agree with the referee’s comment and Supplementary Fig. 3e has now been moved to the main manuscript as Fig. 2d. Meanwhile, we have also added the following sentence to the first paragraph on page 4

“*In addition, Hall measurements was performed on a 3-L device A5, as shown in Fig. 2d, quantized Hall resistance R_{xy} plateaus are resolved at the longitudinal resistance R_{xx} minimum, attesting to the quantum Hall nature of the observed octets. Additional transport data on A5 can be found in Supplementary Fig. 3b-d.*”